# Bandits Dueling on Partially Ordered Sets

**Julien Audiffren**
CMLA
ENS Paris-Saclay, CNRS
Université Paris-Saclay, France
`julien.audiffren@gmail.com`

**Liva Ralaivola**
Lab. Informatique Fondamentale de Marseille
CNRS, Aix Marseille University
Institut Universitaire de France
F-13288 Marseille Cedex 9, France
`liva.ralaivola@lif.univ-mrs.fr`

## Abstract

We address the problem of dueling bandits defined on partially ordered sets, or *posets*. In this setting, arms may not be comparable, and there may be several (incomparable) optimal arms. We propose an algorithm, `UnchainedBandits`, that efficiently finds the set of optimal arms —the *Pareto front*— of any poset even when pairs of comparable arms cannot be a priori *distinguished* from pairs of incomparable arms, with a set of minimal assumptions. This means that `UnchainedBandits` does not require information about comparability and can be used with limited knowledge of the poset. To achieve this, the algorithm relies on the concept of decoys, which stems from social psychology. We also provide theoretical guarantees on both the regret incurred and the number of comparison required by `UnchainedBandits`, and we report compelling empirical results.

## 1 Introduction

Many real-life optimization problems pose the issue of dealing with a few, possibly conflicting, objectives: think for instance of the choice of a phone plan, where a right balance between the price, the network coverage/type, and roaming options has to be found. Such *multi-objective optimization problems* may be studied from the multi-armed bandits perspective (see e.g. Drugan and Nowe [2013]), which is what we do here from a *dueling bandits* standpoint.

**Dueling Bandits on Posets.** Dueling bandits [Yue et al., 2012] pertain to the $K$-armed bandit framework, with the assumption that there is no direct access to the reward provided by any single arm and the only information that can be gained is through the simultaneous pull of two arms: when such a pull is performed the agent is informed about the winner of the *duel* between the two arms. We extend the framework of dueling bandits to the situation where there are pairs of arms that are not comparable, that is, we study the case where there might be no natural order that could help decide the winner of a duel—this situation may show up, for instance, if the (hidden) values associated with the arms are multidimensional, as is the case in the multi-objective setting mentioned above. The notion of incomparability naturally links this problem with the theory of posets and our approach take inspiration from works dedicated to selecting and sorting on posets [Daskalakis et al., 2011].

**Chasing the Pareto Front.** In this setting, the best arm may no longer be unique, and we consider the problem of identifying among all available $K$ arms the set of *maximal* incomparable arms, or the *Pareto front*, with minimal regret. This objective significantly differs from the usual objective of dueling bandit algorithms, which aim to find one optimal arm—such as a Condorcet winner, a Copeland winner or a Borda winner—and pull it as frequently as possible to minimize the regret. Finding the entire Pareto front (denoted $\mathcal{P}$) is more difficult, but pertains to many real-world applications. For instance, in the discussed phone plan setting, $\mathcal{P}$ will contain both the cheapest plan and the plan offering the largest coverage, as well as any non dominated plan in-between; therefore, every customer may then find a a suitable plan in $\mathcal{P}$ in accordance with her personal preferences.

**Key: Indistinguishability.** In practice, the incomparability information might be difficult to obtain. Therefore, we assume the underlying incomparability structure, is *unknown* and *inaccessible.* A pivotal issue that arises is that of *indistinguishability.* In the assumed setting, the pull of two arms that are comparable and that have close values—and hence a probability for either arm to win a duel close to 0.5—is essentially driven by the same random process, i.e. an unbiased coin flip, as the draw of two arms that are not comparable. This induces the problem of indistinguishability: that of deciding from pulls whether a pair of arms is incomparable or is made of arms of similar strengths.

**Contributions.** Our main contribution, the `UnchainedBandits` algorithm, implements a strategy based on a *peeling* approach (Section 3). We show that `UnchainedBandits` can find a nearly optimal approximation of the the set of optimal arms of $\mathcal{S}$ with probability at least $1-\delta$ while incurring a regret upper bounded by $\mathcal{R} \leq \mathcal{O}\left(K\mathbf{width}(S)\log\frac{K}{\delta}\sum_{i,i\notin\mathcal{P}}\frac{1}{\Delta_i}\right)$, where $\Delta_i$ is the regret associated with arm $i$, $K$ the size of the poset and $\mathbf{width}(\mathcal{S})$ its *width*, and that this regret is essentially optimal. Moreover, we show that with little additional information, `Unchained-Bandits` can recover the exact set of optimal arms, and that even when no additional information is available, `UnchainedBandits` can recover $\mathcal{P}$ by using *decoy* arms—an idea stemming from social psychology, where decoys are used to lure an agent (e.g., a customer) towards a specific good/action (e.g. a product) by presenting her a choice between the targetted good and a degraded version of it (Section 4). Finally, we report results on the empirical performance of our algorithm in different settings (Section 5).

**Related Works.** Since the seminal paper of Yue et al. [2012] on Dueling Bandits, numerous works have proposed settings where the total order assumption is relaxed, but the existence of a Condorcet winner is assumed [Yue and Joachims, 2011, Ailon et al., 2014, Zoghi et al., 2014, 2015b]. More recent works [Zoghi et al., 2015a, Komiyama et al., 2016], which envision bandit problems from the social choice perspective, pursue the objective of identifying a Copeland winner. Finally, the works closest to our partial order setting are [Ramamohan et al., 2016] and [Dudík et al., 2015]. The former proposes a general algorithm which can recover many sets of winners—including the uncovered set, which is akin to the Pareto front; however, it is assumed the problems do not contain ties while in our framework, any pair of incomparable arms is encoded as a tie. The latter proposes an extension of dueling bandits using contexts, and introduces several algorithms to recover a Von Neumann winner, i.e. a mixture of arms that is better that any other—and in our setting, any mixture of arms from the Pareto front is a Von Neumann winner. It is worth noting that the aforementioned works aim to identify a single winner, either Condorcet, Copeland or Von Neumann. This is significantly different from the task of identifying the *entire* Pareto front. Moreover, the incomparability property is not addressed in previous works; if some algorithms may still be applied if incomparability is encoded as a tie, they are not designed to fully use this information, which is reflected by their performances in our experiments. Moreover, our lower bound illustrates the fact that our algorithm is essentially optimal for the task of identifying the Pareto front. Regarding decoys, the idea originates from social psychology; they introduce the idea that the introduction of strictly dominated alternatives may influence the perceived value of items. This has generated an abundant literature that studied decoys and their uses in various fields (see e.g. Tversky and Kahneman [1981], Huber et al. [1982], Ariely and Wallsten [1995], Sedikides et al. [1999]). From the computer science literature, we may mention the work of Daskalakis et al. [2011], which addresses the problem of selection and sorting on posets and provides relevant data structures and accompanying analyses.

## 2 Problem: Dueling Bandits on Posets

We here briefly recall base notions and properties at the heart of our contribution.

**Definition 2.1** (Poset). *Let $\mathcal{S}$ be a set of elements. $(\mathcal{S}, \succcurlyeq)$ is a* partially ordered set *or* poset *if $\succcurlyeq$ is a partial reflexive, antisymmetric and transitive binary relation on $\mathcal{S}$.*

**Transitivity relaxation.** Recent works on dueling bandits (see e.g. Zoghi et al. [2014]) have shown that the transitivity property is not required for the agent to successfully identify the maximal element (in that cas,e the Condorcet winner), if it is assumed to exists. Similarly, most of the results we provide do not require transitivity. In the following, we dub *social poset* a transitivity-free poset, i.e. a partial binary relation which is solely reflexive and antisymmetric.

**Remark 2.2.** *Throughout, we will use $\mathcal{S}$ to denote indifferently the set $\mathcal{S}$ or the social poset $(\mathcal{S}, \succcurlyeq)$, the distinction being clear from the context. We make use of the additional notation: $\forall a, b \in \mathcal{S}$*

- $a \parallel b$ *if* $a$ *and* $b$ *are* incomparable *(neither $a \succcurlyeq b$ nor $b \succcurlyeq a$);*
- $a \succ b$ *if $a \succcurlyeq b$ and $a \neq b$;*

**Definition 2.3** (Maximal element and Pareto front). *An element $a \in S$ is a* maximal element *of $S$ if $\forall b \in S$, $a \succcurlyeq b$ or $a \parallel b$. We denote by $\mathcal{P}(S) \doteq \{a : a \succcurlyeq b$ or $a \parallel b, \forall b \in S\}$, the set of maximal elements or Pareto front of the social poset.*

Similarly to the problem of the existence of a Condorcet winner, $\mathcal{P}$ might be empty for social poset (in with posets there always is at least one maximal element). In the following, we assume that $|\mathcal{P}| > 0$. The notions of chain and antichain are key to identify $\mathcal{P}$.

**Definition 2.4** (Chain, Antichain, Width and Height). *$\mathcal{C} \subset S$ is a* chain *(resp. an* antichain*) if $\forall a, b \in \mathcal{C}$, $a \succcurlyeq b$ or $b \succcurlyeq a$ (resp. $a \parallel b$). $\mathcal{C}$ is* maximal *if $\forall a \in S \setminus \mathcal{C}$, $\mathcal{C} \cup \{a\}$ is not a chain (resp. an antichain). The* height *(resp.* width*) of $S$ is the size of its longest chain (resp. antichain).*

**K-armed Dueling Bandit on posets.** The $K$-armed dueling bandit problem on a social poset $S = \{1, \ldots, K\}$ of arms might be formalized as follows. For all maximal chains $\{i_1, \ldots, i_m\}$ of $m$ arms there exist a family $\{\gamma_{i_p i_q}\}_{1 \leq p, q \leq m}$ of parameters such that $\gamma_{ij} \in (-1/2, 1/2)$ and the pull of a pair $(i_p, i_q)$ of arms from the same chain is the independent realization of a Bernoulli random variable $B_{i_p i_q}$ with expectation $\mathbb{E}(B_{i_p i_q}) = 1/2 + \gamma_{i_p i_q}$, where $B_{i_p i_q} = 1$ means that $i$ is the winner of the duel between $i$ and $j$ and conversely (note that: $\forall i, j, \ \gamma_{ji} = -\gamma_{ij}$). In the situation where the pair of arms $(i_p, i_q)$ selected by the agent corresponds to arms such that $i_p \parallel i_q$, a pull is akin to the toss of an unbiased coin flip, that is, $\gamma_{i_p i_q} = 0$. This is summarized by the following assumption:

**Assumption 1** (Order Compatibility). *$\forall i, j \in S$, $(i \succ j)$ if and only if $\gamma_{ij} > 0$.*

**Regret on posets.** In the total order setting, the regret incurred by pulling an arm $i$ is defined as the difference between the best arm and arm $i$. In the poset framework, there might be multiple 'best' arms, and we chose to define regret as the *maximum* of the difference between arm $i$ and the best arm comparable to $i$. Formally, the regret $\Delta_i$ is defined as :

$$\Delta_i = \max\{\gamma_{ji}, \forall j \in \mathcal{P} \text{ such that } j \succcurlyeq i\}.$$

We then define the regret incurred by comparing two arms $i$ and $j$ by $\Delta_i + \Delta_j$. Note the regret of a comparison is zero if and only if the agent is comparing two elements of the Pareto front.

**Problem statement.** The problem that we want to tackle is to identify the Pareto front $\mathcal{P}(S)$ of $S$ as efficiently as possible. More precisely, we want to devise pulling strategies such that for any given $\delta \in (0, 1)$, we are ensured that the agent is capable, with probability $1 - \delta$ to identify $\mathcal{P}(S)$ with a controlled number of pulls *and* a bounded regret.

$\varepsilon$**-indistinguishability.** In our model, we assumed that if $i \parallel j$, then $\gamma_{ij} = 0$: if two arms cannot be compared, the outcome of the their comparison will only depend on circumstances independent from the arms (like luck or personal tastes). Our encoding of such framework makes us assume that when considered over many pulls, the effects of those circumstances cancel out, so that no specific arm is favored, whence $\gamma_{ij} = 0$. The limit of this hypothesis and the robustness of our results when not satisfied are discussed in Section 5.

This property entails the problem of indistinguishability evoked previously. Indeed, given two arms $i$ and $j$, regardless of the number of comparisons, an agent may never be sure if either the two arms are very close to each other ($\gamma_{ij} \approx 0$ and i and j are comparable) or if they are not comparable ($\gamma_{ij} = 0$). This raises two major difficulties. First, any empirical estimation $\hat{\gamma}_{ij}$ of $\gamma_{ij}$ being close to zero is no longer a sufficient condition to assert that $i$ and $j$ have similar values; insisting on pulling the pair $(i, j)$ to decide whether they have similar value may incur a very large regret if they are incomparable. Second, it is impossible to ensure that two elements are incomparable—therefore, identifying the exact Pareto set is intractable if no additional information is provided. Indeed, the agent might never be sure if the candidate set no longer contains unnecessary additional elements—i.e. arms very close to the real maximal elements but nonetheless dominated. This problem motivates the following definition, which quantifies the notion of indistinguishability:

**Definition 2.5** ($\varepsilon$-indistinguishability). *Let $a, b \in S$ and $\varepsilon > 0$. $a$ and $b$ are $\varepsilon$-indistinguishable, noted $a \parallel_\varepsilon b$, if $|\gamma_{ab}| \leq \varepsilon$.*

As the notation $\parallel_\varepsilon$ implies, the $\varepsilon$-indistinguishability of two arms can be seen as a weaker form of incomparability, and note that as $\varepsilon$-decreases, previously indistinguishable pairs of arms become dis-

---

**Algorithm 1** Direct comparison

---

**Given** $(\mathcal{S}, \succ)$ a social poset, $\delta, \varepsilon > 0$, $a, b \in \mathcal{S}$
**Define** $p_{ab}$ the average number of victories of a over b and $I_{ab}$ its $1 - \delta$ confidence interval.
**Compare** $a$ and $b$ until $|I_{ab}| < \varepsilon$ or $0.5 \notin I_{ab}$.
**return** $a \parallel_\varepsilon b$ if $|I_{ab}| < \varepsilon$, else $a \succ b$ if $p_{ab} > 0.5$, else $b \succ a$.

---

---

**Algorithm 2** `UnchainedBandits`

---

**Given** $\mathcal{S} = \{s_1, \ldots, s_K\}$ a social poset, $\delta > 0$, $N > 0$, $(\varepsilon_t)_{t=1}^N \in \mathbb{R}_+^N$
**Define** Set $\mathcal{S}_0 = \mathcal{S}$. Maintain $\hat{p} = (\hat{p}_{ij})_{i,j=1}^K$ the average number of victories of $i$ against $j$ and
$I = (I_{ij})_{i,j=1}^K = \min\left(\sqrt{\frac{\log(NK^2/\delta)}{2n_{ij}}}, 1\right)$ the corresponding $1 - \delta/NK^2$ confidence interval.
**Peel** $\widehat{\mathcal{P}}$**:**      **for** $t = 1$ **to** $N$ **do**     $\mathcal{S}_{t+1} =$ `UBSRoutine` $(\mathcal{S}_t, \varepsilon_t, \delta/N, \mathcal{A} =$ Algorithm 1$)$.
**return** $\widehat{\mathcal{P}} = \mathcal{S}_{N+1}$

---

tinguishable, and the only $0-$indistinguishable pair of arms are the incomparable pairs. The classical notions of a poset related to incomparability can easily be extended to fit the $\varepsilon$-indistinguishability:

**Definition 2.6** ($\varepsilon$-antichain, $\varepsilon$-width and $\varepsilon$-approximation of $\mathcal{P}$)**.** *Let $\varepsilon > 0$. $\mathcal{C} \subset \mathcal{S}$ is an $\varepsilon$-antichain if $\forall a \neq b \in \mathcal{C}$, we have $a \parallel_\varepsilon b$. Additionally, $\mathcal{P}' \subset \mathcal{S}$ is an $\varepsilon$-approximation of $\mathcal{P}$ (noted $\mathcal{P}' \in \mathcal{P}_\varepsilon$) if $\mathcal{P} \subset \mathcal{P}'$ and $\mathcal{P}'$ is an $\varepsilon$-antichain. Finally, $\mathbf{width}_\varepsilon(\mathcal{S})$ is the size of the largest $\varepsilon$-antichain of $\mathcal{S}$.*

**Features of $\mathcal{P}_\varepsilon$.** While the Pareto front is always unique, it might possess multiple $\varepsilon$-approximations. The interest of working with $\mathcal{P}_\varepsilon$ is threefold: i) to find an $\varepsilon$-approximation of $\mathcal{P}$, the agent only has to remove the elements of $\mathcal{S}$ which are not $\varepsilon$-indistinguishable from $\mathcal{P}$; thus, if $\mathcal{P}$ cannot be recovered in the partially observable setting, an $\varepsilon$-approximation of $\mathcal{P}$ can be obtained; ii) any set in $\mathcal{P}_\varepsilon$ contains $\mathcal{P}$, so no maximal element is discarded; iii) for any $B \in \mathcal{P}_\varepsilon$ all the elements of $B$ are nearly optimal, in the sense that $\forall i \in B$, $\Delta_i < \varepsilon$. It is worth noting that $\varepsilon$-approximations of $\mathcal{P}$ may structurally differ from $\mathcal{P}$ in some settings, though. For instance, if $\mathcal{S}$ includes an isolated cycle, an $\varepsilon$-approximation of the Pareto front may contain elements of the cycle and in such case, approximating the Pareto front using $\varepsilon$-approximation may lead to counterintuitive results.

Finding an $\varepsilon$-approximation of $\mathcal{P}$ is the focus of the next subsection.

## 3 Chasing $\mathcal{P}_\varepsilon$ with `UnchainedBandits`

### 3.1 Peeling and the `UnchainedBandits` Algorithm

While deciding if two arms are incomparable or very close is intractable, the agent is able to find if two arms $a$ and $b$ are $\varepsilon$-indistinguishable, by using for instance the *direct comparison* process provided by Algorithm 1. Our algorithm, `UnchainedBandits`, follows this idea to efficiently retrieve an $\varepsilon$-approximation of the Pareto front. It is based on a peeling technique: given $N > 0$ and a decreasing sequence $(\varepsilon_t)_{1 \leq t \leq N}$ it computes and refines an $\varepsilon_t$-approximation $\widehat{\mathcal{P}}_t$ of the Pareto front, using `UBSRoutine` (Algorithm 3), which considers $\varepsilon_t$-indistinguishable arms as incomparable.

**Peeling $\mathcal{S}$.** Peeling provides a way to control the time spent on pulling indistinguishable arms, and it is used to upper bound the regret. Without peeling, i.e. if the algorithm were directly called with $\varepsilon_N$, the agent could use a number of pulls proportional to $1/\varepsilon_N^2$ trying to distinguish two incomparable arms, even though one of them is a regret inducing arm (e.g. an arm $j$ with a large $|\gamma_{i,j}|$ for some $i \in \mathcal{P}$). The peeling strategy ensures that inefficient arms are eliminated in early epochs, before the agent can focus on the remaining arms with an affordable larger number of comparisons.

**Algorithm subroutine.** At each epoch, `UBSRoutine` (Algorithm 3), called on $\mathcal{S}_t$ with parameter $\varepsilon > 0$ and $\delta > 0$, works as follows. It chooses a single initial *pivot*—an arm to which other arms are compared—and successively examines all the elements of $\mathcal{S}_t$. The examined element $p$ is compared to all the pivots (the current pivot and the previously collected ones), using Algorithm 1 with parameters $\varepsilon$ and $\delta/K^2$. Each pivot that is dominated by $p$ is removed from the pivot set. If after being compared to all the pivots, $p$ has not been dominated, it is added to the pivot set. At the end, the set of remaining pivots is returned.

---
**Algorithm 3** `UBSRoutine`
---
**Given** $\mathcal{S}_t$ a social poset, $\varepsilon_t > 0$ a precision criterion, $\delta'$ an error parameter

**Initialisation** Choose $p \in \mathcal{S}_t$ at random. Define $\widehat{\mathcal{P}} = \{p\}$ the set of pivots.

**Construct** $\widehat{\mathcal{P}}$

**for** $c \in \mathcal{S}_t \setminus \{p\}$ **do**

    **for** $c' \in \widehat{\mathcal{P}}$, compare $c$ and $c'$ using Algorithm 1 with $(\delta = \delta'/|S_t|^2, \varepsilon = \varepsilon_t)$.

        $\forall c' \in \widehat{\mathcal{P}}$, such that $c \succ c'$, remove $c'$ from $\widehat{\mathcal{P}}$

    **if** $\forall c' \in \widehat{\mathcal{P}}$, $c \parallel_{\varepsilon_t} c'$ **then** add $c$ to $\widehat{\mathcal{P}}$

**return** $\hat{\mathcal{P}}$

---

**Reuse of informations.** To optimize the efficiency of the peeling process, `UnchainedBandits` reuses previous comparison results: the empirical estimates $p_{ab}$ and the corresponding confidence intervals $I_{ab}$ are initialized using the statistics collected from previous pulls of $a$ and $b$.

### 3.2 Regret Analysis

In this part, we focus on geometrically decreasing peeling sequence, i.e. $\exists \beta > 0$ such that $\varepsilon_t = \beta^t \quad \forall n \geq 0$. We now introduce the following Theorem[1] which gives an upper bound on the regret incurred by `UnchainedBandits`.

**Theorem 1.** *Let $\mathcal{R}$ be the regret generated by Algorithm 2 applied on $\mathcal{S}$ with parameters $\delta$, $N$ and with a decreasing sequence $(\varepsilon_t)_{t=1}^{N}$ such that $\varepsilon_t = \beta^t$, $\forall t \geq 0$. Then with probability at least $1 - \delta$,* `UnchainedBandits` *successfully returns $\hat{\mathcal{P}} \in \mathcal{P}_{\varepsilon_N}$ after at most $T$ comparisons, with*

$$T \leq \mathcal{O}\left(K\mathbf{width}_{\varepsilon_N}(\mathcal{S})\log(NK^2/\delta)/\varepsilon_N^2\right) \tag{1}$$

$$\mathcal{R} \leq \frac{2K}{\beta^2}\log\left(\frac{2NK^2}{\delta}\right)\sum_{i=1}^{K}\frac{1}{\Delta_i} \tag{2}$$

The $1/\beta^2$ reflects the fact that a careful peeling, i.e. $\beta$ close to 1, is required to avoid unnecessary expensive (regret-wise) comparisons: this prevents the algorithm from comparing two incomparable—yet severely suboptimal—arms for an extended period of time. Conversely, for a given approximation accuracy $\varepsilon_N = \varepsilon$, $N$ increases as $-1/\log \beta$, since $\beta^N = \varepsilon$, which illustrates the fact that unnecessary peeling, i.e. peeling that do not remove any arms, lead to a slightly increased regret. In general, $\beta$ should be chosen close to 1 (e.g. 0.95), as the advantages tend to surpass the drawbacks—unless additional information about the poset structure are known.

**Influence of the complexity of $\mathcal{S}$.** In the bounds of Theorem 1, the complexity of $\mathcal{S}$ influences the result through its total size $|\mathcal{S}| = K$ and its width. One of the features of `UnchainedBandits` is that the dependency in $\mathcal{S}$ in Theorem 1 is $|\mathcal{S}|\mathbf{width}(\mathcal{S})$ and not $|\mathcal{S}|^2$. For instance, if $\mathcal{S}$ is actually equipped with a total order, then $\mathbf{width}(\mathcal{S}) = 1$ and we recover the best possible dependency in $|\mathcal{S}|$—which is highlighted by the lower bound (see Theorem 2).

**Comparison Lower Bound.** We will now prove that the previous result is nearly optimal in order. Let $\mathcal{A}$ denotes a dueling bandit algorithm on hidden posets. We first introduce the following Assumption:

**Assumption 2.** $\forall K > W \in \mathbb{N}_*^+$, *for all $\delta > 0$, $1/8 > \varepsilon > 0$, for any poset $\mathcal{S}$ such that $|\mathcal{S}| \leq K$ and $\max(|\mathcal{P}_\varepsilon(\mathcal{S})|) \leq W$, $\mathcal{A}$ identify an $\varepsilon$-approximation of the Pareto front $\mathcal{P}_\varepsilon$ of $\mathcal{S}$ with probability at least $1 - \delta$ with at most $T_{\mathcal{A}}^{\delta,\varepsilon}(K, W)$ comparisons.*

**Theorem 2.** *Let $\mathcal{A}$ be a dueling bandit algorithm satisfying Assumption 2. Then for any $\delta > 0$, $1/8 > \varepsilon > 0$, $K$ and $W$ two positive integers such that $K > W > 0$, there exists a poset $\mathcal{S}$ such that $|\mathcal{S}| = K$, $\mathbf{width}(S) = |\mathcal{P}(\mathcal{S})| = W$, $\max(|\mathcal{P}_\varepsilon(\mathcal{S})|) \leq W$ and*

$$\mathbb{E}\left(T_{\mathcal{A}}^{\delta,\varepsilon}(K, W)|\mathcal{A}(\mathcal{S}) = \mathcal{P}(\mathcal{S})\right) \geq \widetilde{\Theta}\left(KW\frac{\log(1/\delta)}{\varepsilon^2}\right).$$

The main discrepancy between the usual dueling bandit upper and lower bounds for regret is the $K$ factor (see e.g. [Komiyama et al., 2015]) and ours is arguably the $K$ factor. It is worth noting that

**Algorithm 4** Decoy comparison

---

$\qquad$**Given** $(\mathcal{S}, \succ)$ a poset, $\delta, \Delta > 0$, $a, b \in \mathcal{S}$

$\qquad$**Initialisation** Create $a', b'$ the respective $\Delta$- decoy of $a, b$. Maintains $p_{ab}$ the average number of victory of a over b and $I_{ab}$ its $1 - \delta/2$ confidence interval,

$\qquad$**Compare** $a$ and $b'$, $b$ and $a'$, until $\max(|I_{ab'}|, |I_{ba'}|) < \Delta$ or $p_{ab'} > 0.5$ or $p_{a'b} > 0.5$.

$\qquad$**return** $a \parallel_\varepsilon b$ if $\max(|I_{ab'}|, |I_{ba'}|) < \Delta$, else $a \succ b$ if $p_{ab'} > 0.5$, else $b \succ a$.

---

this additional complexity is directly related to the goal of finding the *entire* Pareto front, as can be seen in the proof of Theorem 2 (see Supplementary).

## 4 Finding $\mathcal{P}$ using Decoys

In this section, we discuss several methods to recover the exact Pareto front from an $\varepsilon$-approximation, when $\mathcal{S}$ is a poset. First, note that $\mathcal{P}$ can be found if additional information on the poset is available. For instance, if a lower bound $c > 0$ on the minimum distance of any arm to the Pareto set—defined as $d(\mathcal{P}) = \min\{\gamma_{ij}, \forall i \in \mathcal{P}, j \in \mathcal{S} \setminus \mathcal{P}$, such that $i \succ j\}$—is known, then since $\mathcal{P}_c = \{\mathcal{P}\}$, Un-chainedBandits used with $\varepsilon_N = c$ will produce the Pareto front of $\mathcal{S}$. Alternatively, if the size $k$ of the Pareto front is known, $\mathcal{P}$ can be found by peeling $\mathcal{S}_t$ until it achieves the desired size. This can be achieved by successively calling UBSRoutine with parameters $\mathcal{S}_t$, $\varepsilon_t = \beta^t$, and $\delta_t = 6\delta/\pi^2 t^2$, and by stopping as soon as $|\mathcal{S}_t| = k$.

This additional information may be unavailable in practice, so we propose an approach which does not rely on external information to solve the problem at hand. We devise a strategy which rests on the idea of *decoys*, that we now fully develop. First, we formally define decoys for posets, and we prove that it is a sufficient tool to solve the incomparability problem (Algorithm 4). We also present methods for building those decoys, both for the purely formal model of posets and for real-life problems. In the following, $\Delta$ is a strictly positive real number.

**Definition 4.1** ($\Delta$-decoy). *Let $a \in \mathcal{S}$. Then $b \in \mathcal{S}$ is said to be a $\Delta$-decoy of $a$ if :*

 1. *$a \succcurlyeq b$ and $\gamma_{a,b} \geq \Delta$;*
 2. *$\forall c \in \mathcal{S}, a \parallel c$ implies $b \parallel c$;*
 3. *$\forall c \in \mathcal{S}$ such that $c \succcurlyeq a$, $\gamma_{c,b} \geq \Delta$.*

The following proposition illustrates how decoys can be used to assess incomparability.

**Proposition 4.2** (Decoys and incomparability). *Let $a$ and $b \in \mathcal{S}$. Let $a'$ (resp. $b'$) be a $\Delta$-decoy of $a$ (resp. $b$). Then $a$ and $b$ are comparable if and only if $\max(\gamma_{b,a'}, \gamma_{a,b'}) \geq \Delta$.*

Algorithm 4 is derived from this result. The next proposition, which is an immediate consequence of Proposition 4.2, gives a theoretical guarantee on its performance.

**Proposition 4.3.** *Algorithm 4 returns the correct incomparability result with probability at least $1 - \delta$ after at most $T$ comparisons, where $T = 4\log(4/\delta)/\Delta^2$.*

**Adding decoys to a poset.** A poset $\mathcal{S}$ may not contain all the necessary decoys. To alleviate this, the following proposition states that it is always possible to add relevant decoys to a poset.

**Proposition 4.4** (Extending a poset with a decoy). *Let $(\mathcal{S}, \succcurlyeq, \gamma)$ be a dueling bandit problem on a poset $\mathcal{S}$ and $a \in \mathcal{S}$. Define $a', \mathcal{S}', \succ', \gamma'$ as follows:*

 • *$\mathcal{S}' = \mathcal{S} \cup \{a'\}$*
 • *$\forall b, c \in \mathcal{S}, b \succcurlyeq c$ i.f.f. $b \succcurlyeq' c$ and $\gamma'_{b,c} = \gamma_{b,c}$*
 • *$\forall b \in \mathcal{S}$, if $b \succcurlyeq a$ then $b \succcurlyeq a'$ and $\gamma'_{b,a'} = \max(\gamma_{b,a}, \Delta)$. Otherwise, $b \parallel a'$.*

*Then $(\mathcal{S}', \succcurlyeq', \gamma')$ defines a dueling bandit problem on poset, $\gamma'_{|S} = \gamma$, and $a'$ is a $\Delta$-decoy of $a$.*

Note that the addition of decoys in a poset does not disqualify previous decoys, so that this proposition can be used iteratively to produce the required number of decoys.

**Decoys in real-life.** The intended goal of a decoy $a'$ of $a$ is to have at hand an arm that is known to be lesser than $a$. Creating such a decoy in real-life can be done by using a degraded version of $a$: for the case of an item in a online shop, a decoy can be obtained by e.g. increasing the price. Note that while for large values of the $\Delta$ parameter of the decoys Algorithm 4 requires less comparisons (see

Table 1: Comparison between the five films with the highest average scores (bottom line) and the five films of the computed $\varepsilon$-pareto set (top line).

| Pareto Front | Pulp Fiction | Fight Club | Shawshank Redemption | The Godfather | Star Wars Ep. V |
|---|---|---|---|---|---|
| Top Five | Pulp Fiction | Usual Suspect | Shawshank Redemption | The Godfather | The Godfather II |

Proposition 4.3), in real-life problems, the second point of Definition 4.1 tends to become false: the new option is actually so worse than the original that the decoy becomes comparable (and inferior) to *all* the other arms, including previously non comparable arms (example: if the price becomes absurd). In that case, the use of decoys of arbitrarily large $\Delta$ can lead to erroneous conclusions about the Pareto front and should be avoided. Given a specific decoy, the problem of estimating $\Delta$ in a real-life problem may seem difficult. However, as decoys are not new—even though the use we make of them here is—a number of methods [Heath and Chatterjee, 1995] have been designed to estimate the quality of a decoy, which is directly related to $\Delta$, and, with limited work, this parameter may be estimated as well. We refer the interested reader to the aforementioned paper (and references therein) for more details on the available estimation methods.

**Using decoys**. As a consequence of Proposition 4.3, Algorithm 3 used with decoys instead of direct comparison and $\varepsilon = \Delta$ will produce the exact Pareto front. But this process can be very costly, as the number of required comparison is proportional to $1/\Delta^2$, even for strongly suboptimal arms. Therefore, our algorithm, `UnchainedBandits`, when combined with decoys, first produces an $\varepsilon$-approximation $\widehat{\mathcal{P}}$ of $\mathcal{P}$ using a peeling approach and direct comparisons before refining it into $\mathcal{P}$ by using Algorithm 3 together with decoys. The following theorems provide guarantees on the performances of this modification of `UnchainedBandits`.

**Theorem 3.** `UnchainedBandits` *applied on* $\mathcal{S}$ *with* $\Delta$ *decoys, parameters* $\delta$,$N$ *and with a decreasing sequence* $(\varepsilon_t)_{t=1}^{N-1}$ *lower bounded by* $\Delta\sqrt{\frac{K}{width(\mathcal{S})}}$, *returns the Pareto front* $\mathcal{P}$ *of* $\mathcal{S}$ *with probability at least* $1 - \delta$ *after at most* $T$ *comparisons, with*

$$T \leq \mathcal{O}\left(K\textbf{width}(\mathcal{S})\log(NK^2/\delta)/\Delta^2\right) \tag{3}$$

**Theorem 4.** `UnchainedBandits` *applied on* $\mathcal{S}$ *with* $\Delta$ *decoys, parameters* $\delta$,$N$ *and with a decreasing sequence* $(\varepsilon_t)_{t=1}^{N-1}$ *such that* $\varepsilon_{N-1} \leq \Delta\sqrt{K}$. *returns the Pareto front* $\mathcal{P}$ *of* $\mathcal{S}$ *with probability at least* $1 - \delta$ *while incurring a regret* $\mathcal{R}$ *such that*

$$\mathcal{R} \leq \frac{2K}{\beta^2} \log\left(\frac{2NK^2}{\delta}\right) \sum_{i=1}^{K} \frac{1}{\Delta_i} + K\textbf{width}(S) \log\left(\frac{2NK^2}{\delta}\right) \sum_{i,\Delta_i < \varepsilon_{N-1}, i \notin \mathcal{P}} \frac{1}{\Delta_i}, \tag{4}$$

Compared to (2), (4) includes an extra term due to the regret incurred by the use of decoys. In this term, the dependency in $S$ is slightly worse ($K\textbf{width}(S)$ instead of $K$). However, this extra regret is limited to arms belonging to an $\varepsilon$-approximation of the Pareto front, i.e. nearly optimal arms.

**Constraints on $\varepsilon$.** Theorem 4 require that $\varepsilon_t \leq \sqrt{K}\Delta$, which implies that only near-optimal arms remain during the decoy step. This is crucial to obtain a reasonable upper bound on the incurred regret, as the number of comparisons using decoys is large ($\approx 1/\Delta^2$) and is the same for every arm, regardless of its regret. Conversely, in Theorem 3—which provides an upper bound on the number of comparisons required to find the Pareto front—the $\varepsilon_t$ are required to be lower bounded. This bound is tight in the (worst-case) scenario where all the arms are $\Delta$-indistinguishable, i.e. peeling cannot eliminate any arm. In that case, any comparison done during the peeling is actually wasted, and the lower bound on $\varepsilon_t$ allows to control the number of comparisons made during the peeling step. In order to satisfy both constraints, $\varepsilon_N$ must be chosen such that $\sqrt{K/\textbf{width}(S)}\Delta \leq \varepsilon_N \leq \sqrt{K}\Delta$. In particular $\varepsilon_N = \sqrt{K}\Delta$ satisfy both condition and does not rely on the knowledge of $\textbf{width}(S)$.

## 5  Numerical Simulations

### 5.1  Simulated Poset

Here, we test `UnchainedBandits` on randomly generated posets of different sizes, widths and heights. To evaluate the performance of `UnchainedBandits`, we compare it to three variants of dueling bandit algorithms which were naively modified to handle partial orders and incomparability:

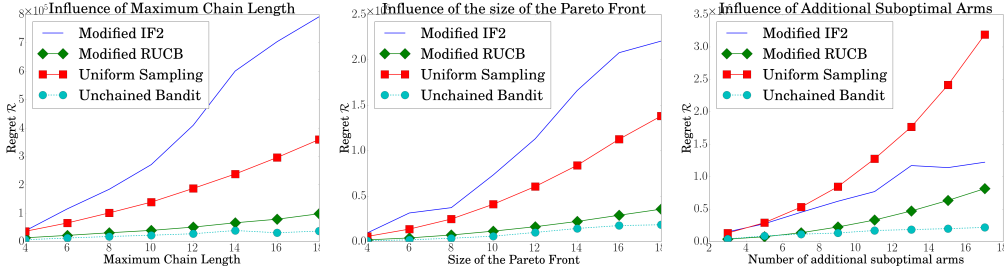

Figure 1: Regret incurred by Modified IF2, Modified RUCB, UniformSampling and UnchainedBandits, when the structure of the poset varies. Dependence on (left:) height, (center:) size of the Pareto front and (right:) addition of suboptimal arms.

1. A simple algorithm, UniformSampling, inspired from the successive elimination algorithm [Even-Dar et al., 2006], which simultaneously compares all possible pairs of arms until one of the arms appears suboptimal, at which point it is removed from the set of selected arms. When only $\Delta$-indistinguishable elements remain, it uses $\Delta$-decoys.

2. A modified version of the single-pivot IF2 algorithm [Yue et al., 2012]. Similarly to the regular IF2 algorithm, the agent maintains a pivot which is compared to every other elements; suboptimal elements are removed and better elements replace the pivot. This algorithm is useful to illustrate consequences of the multi-pivot approach.

3. A modified version of RUCB [Zoghi et al., 2014]. This algorithm is useful to provide a non pivot based perspective.

More precisely, IF2 and RUCB were modified as follows: the algorithms were provided with the additional knowledge of $d(\mathcal{P})$, the minimum gap between one arm of the Pareto front and any other given comparable arm. When during the execution of the algorithm, the empirical gap between two arms reaches this threshold, the arms were concluded to be incomparable. This allowed the agent to retrieve the Pareto front iteratively, one element at a time.

The random posets are generated as follows: a Pareto front of size $p$ is created, and $w$ disjoint chains of length $h - 1$ are added. Then, the top of the chains are connected to a random number of elements of the Pareto front. This creates the structure of the partial order $\succ$. Finally, the exact values of the $\gamma_{ij}$'s are obtained from a uniform distribution, conditioned to satisfy Assumption 1 and to have $d(\mathcal{P}) \geq 0.01$. When needed, $\Delta$-decoys are created according to Proposition 4.4. For each experiment, we changed the value of one parameter, and left the other to their default values ($p = 5$, $w = 2p$, $h = 10$). Additionally, we provide one experiment where we studied the influence of the quality of the arms ($\Delta_i$) on the incurred regret, by adding clearly suboptimal arms[2] to an existing poset. The results are averaged over ten runs, and can be found in reported on Figure 1. By default, we use $\delta = 1/1000$ and $\Delta = 1/100$, $\beta = 0.9$ and $N = \lfloor \log(\sqrt{K}\Delta)/\log\beta) \rfloor$.

**Result Analysis.** While UniformSampling implements a naive approach, it does outperform the modified IF2. This can be explained as in modified IF2, the pivot is constantly compared to all the remaining arms, including all the uncomparable, and potentially strongly suboptimal arms. These uncomparable arms can only be eliminated after the pivot has changed, which can take a large number of comparison, and produces a large regret. UnchainedBandits and modified RUCB produce much better results than UniformSampling and modified IF2, and their advantage increases with the complexity of $\mathcal{S}$. While UnchainedBandits performs better that modified RUCB in all the experiments, it is worth noting that this difference is particularly important when additional suboptimal arms are added. In RUCB, the general idea is roughly to compare the best optimistic arm available to its closest opponent. While this approach works greatly in totally ordered set, in poset it produces a lot of comparisons between an optimal arm $i$ and an uncomparable arm $j$—because in this case $\gamma_{ij} = 0.5$, and $j$ appears to be a close opponent to $i$, even though $j$ can be clearly suboptimal.

## 5.2 MovieLens Dataset

To illustrate the application of `UnchainedBandits` to a concrete example, we used the 20 millions items MovieLens dataset (Harper and Konstan [2015]), which contains movie evaluations. Movies can be seen as a poset, as two movies may be incomparable because they are from different genres (e.g. a horror movie and a documentary). To simulate a dueling bandit on a poset we proceed as follows: we remove all films with less than 50000 evaluations, thus obtaining 159 films, represented as arms. Then, when comparing two arms, we pick at random a user which has evaluated *both* films, and compare those evaluations (ties are broken with an unbiased coin toss). Since the decoy tool cannot be used in an offline dataset, we restrict ourselves to finding an $\varepsilon$-approximation of the Pareto front, with $\varepsilon = 0.05$, and parameters $\beta = 0.9$, $\delta = 0.001$ and $N = \lfloor \log \varepsilon / \log \beta \rfloor = 28$.

Due to the lack of a ground-truth for this experiment, no regret estimation can be provided. Instead, the resulting $\varepsilon$-Pareto front, which contains 5 films, is listed in Table 1, and compared to the five films among the original 159 with the highest average scores. It is interesting to note that three films are present in both list, which reflects the fact that the *best* films in term of average score have a high chance of being in the Pareto Front. However, the films contained in the Pareto front are more diverse in term of genre, which is expected of a Pareto front. For instance, the sequel of the film "The Godfather" has been replaced by a a film of a totally different genre. It is important to remember that `UnchainedBandits` *does not have access to any information about the genre of a film*: its results are based solely on the pairwise evaluation, and this result illustrates the effectiveness of our approach.

**Limit of the uncomparability model.** The hypothesis that i $\parallel$ j $\Rightarrow \gamma_{ij} = 0$ might not always hold true in all real life settings: for instance movies of a niche genre will probably get dominated in users reviews by movies of popular genre—even if they are theoretically incomparable—resulting in their elimination by UnchainedBandit. This might explains why only 5 movies are present in our $\varepsilon$ pareto front. However, even in this case, the algorithm will produce a subset of the Pareto Front, made of uncomparable movies from popular genres. Hence, while the algorithm fails at finding *all* the different genre, it still provides a significant diversity.

# 6    Conclusion

We introduced dueling bandits on posets and the problem of $\varepsilon$-indistinguishability. We provided a new algorithm, `UnchainedBandits`, together with theoretical performance guarantees and compelling experiments to identify the Pareto front. Future work might include the study of the influence of additional hypotheses on the structure of the social poset, and see if some ideas proposed here may carry over to lattices or upper semi-lattices. Additionally, it is an interesting question whether different approaches to dueling bandits, such as Thompson Sampling [Wu and Liu, 2016], could be applied to the partial order setting, and whether results for the von Neumann problem [Balsubramani et al., 2016] can be rendered valid in the poset setting.

## Acknowledgement

We would like to thank the anonymous reviewers of this work for their useful comments, particularly regarding the future work section.

## Footnotes

[1]The complete proof for all our results can be found in the supplementary material.

[2]For this experiment, we say that an arm $j$ is clearly suboptimal if $\exists c \in \mathcal{P}$ s.t. $\gamma_{cj} > 0.15$

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
