[Supplementary Material · unchainedbandit_supplementary.pdf]

# Supplementary Materials for Bandits Dueling on Partially Ordered Sets

## A Appendix : Extended Proofs for social poset

### A.1 Generalization from Poset to social poset.

**Additional notations.** Let $\mathcal{S}, \succ$ be a social poset, $p \in \mathcal{S}$ and $n \in \mathbb{N}$. Let $S' \subset \mathcal{S}$. In the following, we define
$$\Gamma_p^n(S') \doteq \{q \in S', \quad \exists q_1, \ldots q_n \in S' \quad p \succcurlyeq q_1 \succcurlyeq \ldots \succcurlyeq q_n \succcurlyeq q\}.$$

From the definition we immediately have the following Lemmas.

**Lemma A.1** (Characteristics of $\Gamma$). *Let $\mathcal{S}, \succ$ be a social poset, $p \in \mathcal{S}$ and $n \in \mathbb{N}$. Let $S' \subset \mathcal{S}$. We have*

1. $\Gamma_p^n(S') \subset \Gamma_p^{n+1}(S')$

2. *if $S' \subset S'' \subset \mathcal{S}$, then $\Gamma_p^n(S') \subset \Gamma_p^n(S'')$*

3. $\Gamma_p(S') = \bigcup_{n \geq 0} \Gamma_p^n(S')$ *is well defined.*

4. *if $q \in S'$ and $p \succcurlyeq q$, then $\Gamma_q^n(S') \subset \Gamma_p^{n+1}(S')$.*

5. $p \in \mathcal{P}$ *if and only if* $\quad \forall q \neq p, p \notin \Gamma_q(\mathcal{S})$ *if and only if* $\quad \forall q \in \mathcal{S}, p \succcurlyeq q$ or $p \parallel q$.

*Proof.* Immediate by definition. $\square$

**Lemma A.2** ( Poset and $\Gamma$). *Let $\mathcal{S}$ be a social poset. $\mathcal{S}$ is a poset if and only if*
$$\forall p \in \mathcal{S}, \quad \forall S' \subset S \quad \Gamma_p(S') = \Gamma_p^0(S')$$

*Proof.* If $\mathcal{S}$ is a poset, then $\forall q \in \Gamma_p(S'), \exists n \geq 0, \exists q_1, \ldots q_n$ such that $p \succcurlyeq q_1 \succcurlyeq \ldots \succcurlyeq q_n \succcurlyeq q$. Hence $p \succcurlyeq q$ by transitivity, i.e. $q \in \Gamma_p^0(S')$.

Conversely, let $p, q, r$ such that $p \succcurlyeq q$ and $q \succcurlyeq r$. Then $r \in \Gamma_q^0(S) \subset \Gamma_p^1(S) \subset \Gamma_p(S) = \Gamma_p^0(S)$. Hence $p \succcurlyeq r$ $\square$

### A.2 Proof of Proposition 4.2.

*Proof.* Let us assume that $a \succcurlyeq b$. The third point of Definition 4.1 implies that $\gamma_{a,b'} \geq \Delta$, hence $a \succcurlyeq b'$. The rest follows from point 2 of Definition 4.1. $\square$

### A.3 Proof of Proposition 4.3.

*Proof.* This proposition immediately follows from Proposition 4.2 and Hoeffding inequality. $\square$

### A.4 Proof of Proposition 4.4.

*Proof.* The result naturally follows from the definition of a social poset (resp. a Poset) and Definition 4.1. $\square$

### A.5 Proof of Theorem 1.

The proof of Theorem 1 relies on the following intermediate result :

**Proposition A.3.** *Algorithm 3 called at epoch $t$ on $\mathcal{S}_t$ with parameter $\varepsilon_t > 0$, $\delta' > 0$ and $\mathcal{A} =$ Algorithm 1 returns an $\varepsilon_t-$approximation of the Pareto front of $\mathcal{S}_t$ with probability at least $1 - \delta'$ after at most*
$$T \leq 2|\mathcal{S}_t|\textbf{width}_{\varepsilon_t}(\mathcal{S}_t)\log(2|\mathcal{S}_t|^2/\delta')\left(\frac{1}{\varepsilon_t^2} - \mathbf{1}_{t>1}\frac{1}{\varepsilon_{t-1}^2}\right)$$

***additional*** *comparisons, where $\mathbf{1}$ is the indicator function.*

*Proof.* In this setting Algorithm 1 is used for comparisons purpose. We first tackle the case $t = 1$, i.e. the first epoch, since in this case, there is no previous observations, and thus no negative term in the upper bound.

**Case $t = 1$.** The proof for $t = 1$ unfolds similarly to the previous case, with a different invariant.

Let $\mathcal{E}_1$ be the event where during the execution of Algorithm 5, each call to $\mathcal{A}$, Algorithm 1 returns the correct answer:

- $\mathcal{A}(i, j) = i \parallel_\varepsilon j \implies |\gamma_{ij}| < \varepsilon.$
- $\mathcal{A}(i, j) = i \succ j \implies \gamma_{ij} > 0.$
- $\mathcal{A}(i, j) = j \succ i \implies \gamma_{ij} < 0.$

We are going to prove the following invariant for the principal loop of the Algorithm on $\mathcal{E}_1$.

**Invariant:** At the iteration $n$, Let $\mathcal{S}_t^n$ the subset of element of $\mathcal{S}_t$ already considered, $\widehat{\mathcal{P}}^n$ the current set of pivot. Then

$$\forall c' \in \mathcal{S}_t^n \quad \exists c \in \widehat{\mathcal{P}}^n, \quad c' \in \Gamma_c\left(\mathcal{S}_t^n\right) \tag{5}$$
$$\forall c, c' \in \widehat{\mathcal{P}}_t, \quad c \parallel_\varepsilon c' \tag{6}$$

It is easy to see that the invariant is true at the beginning of the algorithm because $\mathcal{S}_t^0 = \widehat{\mathcal{P}}^0$ and $|\widehat{\mathcal{P}}^0| = 1$. Suppose that the invariant is true at the $n$-th iteration. Let $p$ be the new element considered, i.e. $\mathcal{S}_t^{n+1} = \mathcal{S}_t^n \cup \{p\}$.

1. **Case 1.** $\exists q \in \widehat{\mathcal{P}}^n$ s.t. $q \succ p$ and $\gamma_{qp} > \varepsilon$. In this case, $\widehat{\mathcal{P}}^{n+1} \subset \widehat{\mathcal{P}}^n$, hence (6) at iteration $n$ immediatly implies (6) at iteration $n + 1$. Let let p' be a pivot eliminated by $p$, i.e. $p \succ p'$. Since $q \succ p$, we have by Lemma A.2,

    $$\Gamma_{p'}\left(\mathcal{S}_t^n\right) \subset \Gamma_p\left(\mathcal{S}_t^n\right) \subset \Gamma_q\left(\mathcal{S}_t^{n+1}\right).$$

    Hence (5) at iteration $n$ implies (5) at iteration $n + 1$.

2. **Case 2.** $\forall q \in \widehat{\mathcal{P}}^n$, ($p \succ q$ and $\gamma_{pq} > \varepsilon$ ) or $|\gamma_{pq}| < \varepsilon$. Then

    $$\widehat{\mathcal{P}}^{n+1} = \{p\} \cup \widehat{\mathcal{P}}^n \setminus \Gamma_p^0(\widehat{\mathcal{P}}^n),$$

    and is it easy to see that (5) is still true iteration $n + 1$. Now we are going to prove that (6) is still true by RAA. Assume that $\exists q \in \widehat{\mathcal{P}}^{n+1}$ s.t. $q$ is comparable to $p$ and $|\gamma_{qp}| > \varepsilon$. By definition of $\Gamma_p^0$, it implies that $q \succ p$, and the order compatibility of the poset implies that $\gamma_{qp} > \varepsilon$ which contradicts the initial assumption of the case.

After the last iteration n, we have $\mathcal{S}_t^{n+1} = \mathcal{S}_t$, since all the elements have been examined. We now prove by RAA that the invariant implies that $\widehat{\mathcal{P}}^{n+1}$ is an $\varepsilon$-approximation of $\mathcal{P}$. We drop the $n + 1$ in $\widehat{\mathcal{P}}^{n+1}$ for the sake of alleviating the notations.

Now assume that $\mathcal{P} \not\subset \widehat{\mathcal{P}}$ and let $p \in \mathcal{P} \setminus \widehat{\mathcal{P}}$. Hence (5) implies that $\exists q \in \widehat{\mathcal{P}}$ s.t. $p \in \Gamma_q$ which contradicts point 5 of Lemma A.2. Hence $\mathcal{P} \subset \widehat{\mathcal{P}}$.

Now suppose that $\exists q \in \widehat{\mathcal{P}}$ such that $\exists p \in \mathcal{P}$ s.t. $p \succ q$ and $\gamma_{pq} > \varepsilon$. Since $\mathcal{P} \subset \widehat{\mathcal{P}}$, we have $p \in \widehat{\mathcal{P}}$ and thus $\gamma_{pq} > \varepsilon$. contradicts (6). Hence $\widehat{\mathcal{P}}$ is a $\varepsilon$-approximation of $\mathcal{P}$.

A consequence of (6) is that at each step, $\widehat{\mathcal{P}}^n$ is an $\varepsilon$-antichain. Since during the execution of the algorithm all the elements of $\mathcal{S}_t$ are compared to all the element of the current $\widehat{\mathcal{P}}$, the algorithm do at most

$$|\mathcal{S}_t| \max_n |\widehat{\mathcal{P}}^n| \leq |\mathcal{S}_t| \mathbf{width}_\varepsilon(\mathcal{S}_t)$$

comparisons, and as a consequence

$$\mathbb{P}(\mathcal{E}_1^C) \le |\mathcal{S}_t| \mathbf{width}_\varepsilon(\mathcal{S}_t) \frac{\delta}{|\mathcal{S}_t|^2} \le \delta.$$

The upper bound on the number of comparisons results with the same remark combined with the fact that Algorithm 2 uses Hoeffding inequality.

**Case** $1 < t < N$**.**
To conclude, we only need to lower bound the number of previous comparisons that can be reused. Once again, consider the event $\mathcal{E}_1$ be the event where during the execution of `UBSRoutine`, each call to Algorithm 1 returns the correct answer. Let $i$ and $j \in \mathcal{S}_t$ such that $i$ and $j$ are compared at epoch $t$ (i.e. during the call number $t$ of Algorithm 2). Note that $S_t = \widehat{\mathcal{P}}_{t-1}^n$ and let assume without any loss of generality that $i$ was added before $j$ into $\widehat{\mathcal{P}}_{t-1}^n$. Since $i$ is a pivot at the end of the epoch $t-1$, it was compared to all the arm considered after $i$, including $j$.

Since both $i$ and $j$ are pivots at the end of epoch $t-1$, it implies that $i \parallel j$ or $\gamma_{ij} < \varepsilon_{t-1}$. In both cases, Algorithm the algorithm does exactly $\frac{\log(K^2/\delta')}{\varepsilon_{t-1}^2}$ comparisons to reach this conclusion. The result follows from the reuse of information. $\qquad\square$

*Proof of Theorem 1.* First note that if $\mathcal{P}'$ is a $\varepsilon$- approximation of $\mathcal{P}$, then $\mathcal{P} \subset \mathcal{P}'$. Additionally, it is easy to see that if $\mathcal{S}$ is a poset and $\mathcal{P}$ is its Pareto set, then $\forall \mathcal{S}' \subset \mathcal{S}$ such that $\mathcal{P} \subset \mathcal{S}'$, the Pareto front of $\mathcal{S}'$ is $\mathcal{P}$.

Hence, Proposition A.3 implies that with probability at least $1 - N\delta/N = 1 - \delta$, Algorithm 2 returns an $\varepsilon-$ approximation of the pareto front of $\mathcal{S}$. in at most $T$ comparisons, where

$$T \le 2 \sum_{t=1}^{N} |\mathcal{S}_t| \mathbf{width}_{\varepsilon_t}(\mathcal{S}_t) \log(2N|\mathcal{S}_t|^2/\delta) \left( \frac{1}{\varepsilon_t^2} - \mathbf{1}_{t>1} \frac{1}{\varepsilon_{t-1}^2} \right)$$

$$\le 2 \sum_{t=1}^{N-1} \frac{1}{\varepsilon_t^2} \left( |\mathcal{S}_t| \mathbf{width}_{\varepsilon_t}(\mathcal{S}_t) \log(2N|\mathcal{S}_t|^2/\delta) \right.$$
$$\left. - |\mathcal{S}_{t+1}| \mathbf{width}_{\varepsilon_{t+1}}(\mathcal{S}_{t+1}) \log(2N|\mathcal{S}_{t+1}|^2/\delta) \right)$$
$$+ \frac{2}{\varepsilon_N^2} |\mathcal{S}_N| \mathbf{width}_{\varepsilon_N}(\mathcal{S}_N) \log(2N|\mathcal{S}_N|^2/\delta)$$

The upper bound (1) follows from the fact that $|\mathcal{S}_t| \mathbf{width}_{\varepsilon_t}(\mathcal{S}_t) \log(N|\mathcal{S}_t|^2/\delta)$ is decreasing in $t$.

Now we focus on proving the regret upper bound (2). Let $i$ be an arm, and $N_i$ be the last peeling step before $i_p$ is eliminated. If $i$ is not eliminated at the end of the peeling, then we set $N_i = N - 1$. In other words,

$$N_i = \max\{1 \le t \le N, \quad i \in \widehat{\mathcal{P}}_t\}$$
$$= 1 + \max \left( 0, \min \left( \left\lceil \frac{\log(\Delta_i)}{\log(\gamma)} \right\rceil, N \right) \right).$$

Let $j \le N_i$. During the $j$-th phase of peeling, the arm $i$ is compared to at most $|S_j - 1|$ other arms. Hence, with the same argument as in Proposition A.3, we have

$$\mathcal{R} \le 2 \sum_{i=1}^{K} \Delta_i \sum_{t=1}^{N_i} |\mathcal{S}_t| \log(2N|\mathcal{S}_t|^2/\delta) \left( \frac{1}{\varepsilon_t^2} - \mathbf{1}_{t>1} \frac{1}{\varepsilon_{t-1}^2} \right).$$

Now since $\varepsilon_t < \varepsilon_{t-1}$, and $|S_t| \le K$, we have

$$\mathcal{R} \leq 2K \log\left(\frac{2NK^2}{\delta}\right) \sum_{i=1}^{K} \Delta_i \sum_{t=1}^{N_i} \left(\frac{1}{\varepsilon_t^2} - \mathbf{1}_{t>1} \frac{1}{\varepsilon_{t-1}^2}\right)$$

$$\leq 2K \log\left(\frac{2NK^2}{\delta}\right) \sum_{i=1}^{K} \frac{\Delta_i}{\varepsilon_{N_i}^2}.$$

Since by construction, we have $\varepsilon_{t+1} = \gamma \varepsilon_t$, then

$$\mathcal{R} \leq \frac{2K}{\gamma^2} \log\left(\frac{2NK^2}{\delta}\right) \sum_{i=1}^{K} \frac{1}{\Delta_i}$$

hence the conclusion since

$$\gamma \approx \Delta_i^{1/N_i}.$$

$\square$

### A.6 Proof of Theorem 2.

We start by recalling Theorem 2 and its related Assumption 2

**Assumption 2.** $\forall K > W \in \mathbb{N}_*^+$, for any poset $\mathcal{S}$ such that $|\mathcal{S}| \leq K$ and $\max\left(|\mathcal{P}_\varepsilon(\mathcal{S})|\right) \leq W$, for all $\delta > 0$, $1/8 > \varepsilon > 0$ $\mathcal{A}$ identify the $\varepsilon$-approximation of the Pareto front $\mathcal{P}_\varepsilon$ of $\mathcal{S}$ with probability at least $1 - \delta$ with at most $T_{\mathcal{A}}^{\delta,\varepsilon}(K, W)$ comparisons.

To prove this results, we need the following Lemma, derived from Lemma 4.7.2 from [Robert, 1990]

**Lemma A.4** (Lower bound on mistake probability). *Let $\mathcal{S}$ be a poset and $i, j \in \mathcal{S}$ such that $i \parallel_\varepsilon j$. Then*

$$\mathbb{P}\left(p_{i,j}^n > 1 + \varepsilon\right) \geq \frac{1}{\sqrt{2n}} e^{-32n\varepsilon^2}$$

*Proof.* It is easy to see that $p_{i,j}^n$ follows a Binomial law $\mathcal{B}(0.5 + \gamma_{i,j}, n)$. Since $i\parallel_\varepsilon j$, we have $0.5 + \gamma_i > 0.5 - \varepsilon$. Hence

$$\mathbb{P}\left(p_{i,j}^n > 1 + \varepsilon\right) \geq \mathbb{P}\left(\mathcal{B}(0.5 - \varepsilon, n) > 1 + \varepsilon\right)$$

$$\geq \frac{1}{\sqrt{2n}} \exp(-n \log(2) \mathrm{B}(0.5 + \varepsilon \| 0.5 - \varepsilon))$$

where we used Lemma 4.7.2 from [Robert, 1990], and KL denotes the Kullback-Leibler divergence between two Bernoulli random variables. now,

$$\mathrm{B}(0.5 + \varepsilon \| 0.5 - \varepsilon)) = \frac{1 + 2\varepsilon}{2} \log\left(\frac{1 + 2\varepsilon}{1 - 2\varepsilon}\right) + \frac{1 - 2\varepsilon}{2} \log\left(\frac{1 - 2\varepsilon}{1 + 2\varepsilon}\right)$$

$$= 4\varepsilon \log\left(1 + 4\frac{\varepsilon}{1 - 2\varepsilon}\right)$$

$$\leq \frac{16\varepsilon^2}{1 - 2\varepsilon} \leq 32\varepsilon^2$$

$\square$

The interest of Lemma A.4 is the following : assume that during its execution, **A** compare $i$ and $j$ exactly $n$ times before reaching a conclusion on the relation between $i$ and $j$, i.e $i \succ j$, $j \succ i$ or $i\parallel_\varepsilon j$. Then it is easy to see that the previous result translates in a lower bound of the probability of the algorithm reaching the wrong conclusion. This idea is the key component of the proof of Theorem 2, which is inspired by the work of Feige et al. [1994].

*Proof.* (Theorem 2).

Let $\mathcal{S}, \succ, \gamma$ be a dueling bandit problem on a poset defined as follows :

1. $\mathcal{S}$ contains $K$ elements

2. $\mathcal{S}$ is the reunion of $W$ disjoint chains $C_1, \ldots, C_W$.

3. For every pair of elements such that $i \succ j$, then $\gamma_{i,j} = 3\varepsilon/2$

It is easy to see that this poset satisfies $|\mathcal{S}| = K$, $|\mathcal{P}(\mathcal{S})| = W$. Moreover, the only $\varepsilon-$Pareto front of $\mathcal{S}$ is $\mathcal{P}$. Let denote $\mathcal{P} = \{i_1, \ldots, i_W\}$ and let define

$$\tau : \mathcal{S} \to [1, W]$$
$$i \mapsto m, \text{ s.t. } i \in C_m.$$

The construction of $\mathcal{S}$ ensure that $\tau$ is well defined. Now consider the event $\mathcal{E} = \{T_{\mathbf{A}}^{\delta,\varepsilon}(K, W) > T_{max} := KW\frac{\log(1/\delta)}{\varepsilon^2}\}$. If $\mathbb{P}(\mathcal{E}) \geq 1/2$, then the results follows. In the rest of the proof, we consider the case $\mathbb{P}(\mathcal{E}) < 1/2$.

Now, for any $i \in \mathcal{S}, m \in [1, W] \setminus \{\tau(i)\}$, we define the poset $\mathcal{S}_i^m, \succ_i^m$ as follows:

1. $\mathcal{S}$ contains the same element as $\mathcal{S}_i^m$

2. $\forall j, j' \in \mathcal{S}_i^m \setminus \{i\}, (j \succ j' \text{ if and only if } j \succ_i^m j')$.

3. $\forall j \in \mathcal{S}_i^m \setminus C_m, (j \succ i \text{ if and only if } j \succ_i^m i)$.

4. $\forall j \in C_m, i \succ_i^m j$

5. For every pair of elements $j, j' \in \mathcal{S}_i^m$ such that $j \succ_i^m j'$, then $\gamma'_{j,j'} = 3\varepsilon/2$

Finally, for $m = \tau(i)$, define :

1. $\mathcal{S}$ contains the same element as $\mathcal{S}_i^{\tau(i)}$

2. $\forall j \in \mathcal{S}_i^{\tau(i)} \setminus \{i\}, j\|i$

3. For every pair of elements $j, j' \in \mathcal{S}_i^{\tau(i)}$ such that $j \succ_i^{\tau(i)} j'$, then $\gamma'_{j,j'} = 3\varepsilon/2$

It is easy to see that this poset satisfies $|\mathcal{S}_i^m| = K$, $W \geq \mathbf{width}(\mathcal{S}_i^m) \geq W - 1$. Note that by construction in $\mathcal{S}_i^m$ the only modified comparisons are the one between $i$ and the elements of $C_m$. Additionally, $\mathcal{P}(\mathcal{S}) \neq \mathcal{P}(\mathcal{S}_i^m)$. For any $Z \subset \mathcal{S}$, let $n_{i,Z}$ be the number of time the arm $i$ is compared with elements of $Z$ during the execution of $\mathbf{A}$ on $\mathcal{S}$. Additionally, let $\mathbf{A}(\mathcal{S})$ be the result of this computation, and $T$ the total number of comparison used by the execution of $\mathbf{A}$ on $\mathcal{S}$. Then

$$\begin{aligned}
\delta &\geq \mathbb{P}(\mathbf{A}(\mathcal{S}_i^m) \neq \mathcal{P}(\mathcal{S}_i^m)) \\
&\geq \mathbb{E}\left(\frac{1}{\sqrt{2n_{i,C_m}}}(e^{-128\varepsilon^2})^{n_{i,C_m}}|\mathbf{A}(\mathcal{S}) = \mathcal{P}(\mathcal{S})\right)\mathbb{P}\left(\mathbf{A}(\mathcal{S}) = \mathcal{P}(\mathcal{S})\right) \\
&\geq \mathbb{E}\left(\frac{1}{\sqrt{2T}}(e^{-128\varepsilon^2})^{n_{i,C_m}}|\mathbf{A}(\mathcal{S}) = \mathcal{P}(\mathcal{S})\right)(1-\delta),
\end{aligned} \tag{7}$$

where we used Assumption 2. On the other hand, we have that

$$\sum_{i \in \mathcal{S}} \sum_{1 \leq m \leq W} n_{i,C_m} = \sum_{i \in \mathcal{S}} n_{i,\mathcal{S}} = 2T.$$

Hence by using the the inequality of arithmetic and geometric means and (7):

$$KW\frac{\delta}{1-\delta} \geq \frac{1}{1-\delta}\sum_{i\in\mathcal{S}}\sum_{1\leq m\leq W}\mathbb{P}(\mathbf{A}(\mathcal{S}_i^m) \neq \mathcal{P}(\mathcal{S}_i^m))$$

$$\geq \sum_{i\in\mathcal{S}}\sum_{1\leq m\leq W}\mathbb{E}\left(\frac{1}{\sqrt{2T}}(e^{-128\varepsilon^2})^{n_{i,C_m}}|\mathbf{A}(\mathcal{S})=\mathcal{P}(\mathcal{S})\right)$$

$$\geq \sum_{i\in\mathcal{S}}\sum_{1\leq m\leq W}\mathbb{E}\left(\frac{1}{\sqrt{2T}}(e^{-128\varepsilon^2})^{2T/KW}|\mathbf{A}(\mathcal{S})=\mathcal{P}(\mathcal{S})\right)$$

$$\geq KW\mathbb{E}\left(\frac{1}{\sqrt{2T}}(e^{-128\varepsilon^2})^{2T/KW}|\mathbf{A}(\mathcal{S})=\mathcal{P}(\mathcal{S})\right)$$

Now using the previous inequality and Jensen inequality:

$$\log\left(\frac{\delta}{1-\delta}\right) \geq \log\left(\mathbb{E}\left(\frac{1}{\sqrt{2T}}(e^{-128\varepsilon^2})^{2T/KW}|\mathbf{A}(\mathcal{S})=\mathcal{P}(\mathcal{S})\right)\right)$$

$$\geq -\mathbb{E}\left(\frac{256T\varepsilon^2}{KW}|\mathbf{A}(\mathcal{S})=\mathcal{P}(\mathcal{S})\right) - \frac{1}{2}\mathbb{E}\left(\log\left(2T\right)|\mathbf{A}(\mathcal{S})=\mathcal{P}(\mathcal{S})\right)$$

hence the conclusion.

$\square$

# B   Appendix : Extended Proofs for Poset and Decoys

**Proposition B.1.** *Let $S$ let a poset, and $\varepsilon < \min\left(\gamma_{p,q}, \quad p \in \mathcal{P}, q \in \mathcal{S} \setminus \mathcal{P}\right)$. Then $\mathcal{P}_\varepsilon = \{\mathcal{P}\}$*

*Proof.* Let $\widehat{\mathcal{P}} \in \mathcal{P}_\varepsilon$. By Definition 2.6, we have $\mathcal{P} \subset \widehat{\mathcal{P}}$. $\square$

## B.1   Proof of Theorem 3.

This result is a consequence of Theorem 1 and the following intermediate result, whose proof can be found in the Appendix.

**Proposition B.2.** *Algorithm 3 called on $\mathcal{S}_t$ with parameter $\Delta > 0$, $\delta' > 0$ and $\mathcal{A}$ = Algorithm 4 returns the Pareto front of $\mathcal{S}_t$ with probability at least $1 - \delta'$ after at most*

$$T \leq 4|\mathcal{S}_t|\textbf{width}(\mathcal{S}_t)\log(4|\mathcal{S}_t|^2/\delta')/\Delta^2$$

*comparisons.*

First note that if $\mathcal{P}'$ is a $\varepsilon$- approximation of $\mathcal{P}$, then $\mathcal{P} \subset \mathcal{P}'$. Additionally, it is easy to see that if $\mathcal{S}$ is a poset and $\mathcal{P}$ is its Pareto set, then $\forall \mathcal{S}' \subset \mathcal{S}$ such that $\mathcal{P} \subset \mathcal{S}'$, the Pareto front of $\mathcal{S}'$ is $\mathcal{P}$.

Hence, Theorem 1 combined with Proposition B.2 imply that with probability at least $1 - N\delta/N = 1 - \delta$, Algorithm 4 returns the Pareto front of $\mathcal{S}$. in at most T comparisons, where

$$T \leq 2 \sum_{t=1}^{N-1} |\mathcal{S}_t|\mathbf{width}_{\varepsilon_t}(\mathcal{S}_t)\log(2N|\mathcal{S}_t|^2/\delta) \left( \frac{1}{\varepsilon_t^2} - \mathbf{1}_{t>1}\frac{1}{\varepsilon_{t-1}^2} \right)$$
$$+ 4|\mathcal{S}_N|\mathbf{width}(\mathcal{S}_N)\frac{\log(4N|\mathcal{S}_N|^2/\delta)}{\Delta^2}$$
$$\leq 2 \sum_{t=1}^{N-2} \frac{1}{\varepsilon_t^2} \left( |\mathcal{S}_t|\mathbf{width}_{\varepsilon_t}(\mathcal{S}_t)\log(2N|\mathcal{S}_t|^2/\delta) \right.$$
$$\left. -|\mathcal{S}_{t+1}|\mathbf{width}_{\varepsilon_{t+1}}(\mathcal{S}_{t+1})\log(2N|\mathcal{S}_{t+1}|^2/\delta) \right)$$
$$+ \frac{2}{\varepsilon_{N-1}^2}|\mathcal{S}_{N-1}|\mathbf{width}_{\varepsilon_{N-1}}(\mathcal{S}_{N-1})\log(2N|\mathcal{S}_{N-1}|^2/\delta)$$
$$+ 4|\mathcal{S}_N|\mathbf{width}(\mathcal{S}_N)\frac{\log(4N|\mathcal{S}_N|^2/\delta)}{\Delta^2}$$

where the second inequality is obtained by rearranging the sum. Now, by hypothesis we have

$$\varepsilon_t > \varepsilon_{N-1} \geq \Delta\sqrt{\frac{|\mathcal{S}|}{\mathbf{width}(\mathcal{S})}}$$

Hence, since the $|\mathcal{S}_t|\mathbf{width}_{\varepsilon_t}(\mathcal{S}_t)\log(N|\mathcal{S}_t|^2/\delta)$ is decreasing in $t$ we have

$$T \leq 2 \sum_{t=1}^{N-2} \frac{\mathbf{width}(\mathcal{S})}{|\mathcal{S}|\Delta^2} \left( |\mathcal{S}_t|\mathbf{width}_{\varepsilon_t}(\mathcal{S}_t)\log(2N|\mathcal{S}_t|^2/\delta) \right.$$
$$\left. -|\mathcal{S}_{t+1}|\mathbf{width}_{\varepsilon_{t+1}}(\mathcal{S}_{t+1})\log(2N|\mathcal{S}_{t+1}|^2/\delta) \right)$$
$$+ \frac{2\mathbf{width}(\mathcal{S})}{|\mathcal{S}|\Delta^2}|\mathcal{S}_{N-1}|\mathbf{width}_{\varepsilon_{N-1}}(\mathcal{S}_{N-1})\log(2N|\mathcal{S}_{N-1}|^2/\delta)$$
$$+ 4|\mathcal{S}_N|\mathbf{width}(\mathcal{S}_N)\frac{\log(N|\mathcal{S}_N|^2/\delta)}{\Delta^2}$$
$$\leq \frac{2}{\Delta^2}|\mathcal{S}|\frac{\mathbf{width}_{\varepsilon_1}(\mathcal{S})}{|\mathcal{S}|}\mathbf{width}(S)\log(2N|\mathcal{S}|^2/\delta)$$
$$+ 4|\mathcal{S}_N|\mathbf{width}(\mathcal{S}_N)\frac{\log(4N|\mathcal{S}_N|^2/\delta)}{\Delta^2}$$
$$\leq \mathcal{O}\left( K\mathbf{width}(\mathcal{S})\frac{\log(NK^2/\delta)}{\Delta^2} \right)$$

## B.2 Proof of Proposition B.2

In this setting, the arms are compared using decoys. Let $\mathcal{E}_1$ be the event where during the execution of Algorithm UBSRoutine, each call to Algorithm 4 returns the correct answer. We are going to prove the following invariant for the principal loop of the Algorithm on $\mathcal{E}_1$.

**Invariant:** At the iteration $n$, Let $\mathcal{S}_t^n$ the set of element of $\mathcal{S}_t$ already considered, $\widehat{\mathcal{P}}^n$ the current set of pivot. Then

$$\forall c' \in \mathcal{S}_t^n \quad \exists c \in \widehat{\mathcal{P}}^n, \quad c \succcurlyeq c' \tag{8}$$
$$\forall c, c' \in \widehat{\mathcal{P}}_t, \quad c \parallel c' \tag{9}$$

It is easy to see that the invariant is true at the beginning of the algorithm because $\mathcal{S}_t^0 = \widehat{\mathcal{P}}^0$ and $|\widehat{\mathcal{P}}^0| = 1$.

Suppose that the invariant is true at the $n$-th iteration. Let $p$ be the new element considered, i.e. $\mathcal{S}_t^{n+1} = \mathcal{S}_t^n \cup \{p\}$, and define $\Gamma_-^p \doteq \{q \in \widehat{\mathcal{P}}^n, p \succ q\}$

1. **Case 1.** $\exists q \in \widehat{\mathcal{P}}^n$ s.t. $q \succ p$. In this case, $\widehat{\mathcal{P}}^{n+1} = \widehat{\mathcal{P}}^n \setminus \Gamma_-^p$, hence (9) at iteration $n$ immediatly implies (9) at iteration $n + 1$. Since $q \succ p$, we have $\forall q' \in \Gamma_-^p$, we have $q \succ q'$ by transitivity.Hence (8) at iteration n implies (8) at iteration n+1.

2. **Case 2.** $\forall q \in \widehat{\mathcal{P}}^n$, $p \succ q$ or $p \parallel q$. Then

$$\widehat{\mathcal{P}}^{n+1} = \{p\} \cup \widehat{\mathcal{P}}^n \setminus \Gamma_-^p,$$

and is it easy to see that (8) is still true iteration $n + 1$. Now we are going to prove that (9) is still true by RAA. Assume that $\exists q \in \widehat{\mathcal{P}}^{n+1}$ s.t. $q$ is comparable to $p$. By definition of $\Gamma_-^p$, it implies that $q \succ p$, which contradicts the initial assumption of the case.

After the last iteration n, we have $\mathcal{S}_t^{n+1} = \mathcal{S}_t$, since all the elements have been examined. We now prove by RAA that the invariant implies that $\widehat{\mathcal{P}}^{n+1} = \mathcal{P}$. We drop the $n + 1$ in $\widehat{\mathcal{P}}^{n+1}$ for the sake of alleviating the notations.

Suppose that $\widehat{\mathcal{P}} \not\subset \mathcal{P}$ and let $p \in \widehat{\mathcal{P}} \setminus \mathcal{P}$. Since $p \notin \mathcal{P}, \exists q \in \mathcal{P}$ s.t. $q \succ p$. If $q \in \widehat{\mathcal{P}}$, (9) is contracted. Then $q \notin \widehat{\mathcal{P}}$. Hence $q \succ p$ contradicts (8). So $\widehat{\mathcal{P}} \subset \mathcal{P}$.

Now assume that $\mathcal{P} \not\subset \widehat{\mathcal{P}}$ and let $p \in \mathcal{P} \setminus \widehat{\mathcal{P}}$. Since $p \notin \widehat{\mathcal{P}}$, (8) implies that $\exists q \in \widehat{\mathcal{P}}$ s.t. $q \succcurlyeq p$. Since $p \notin \widehat{\mathcal{P}}$ and $q \in \widehat{\mathcal{P}}, q \neq p$ hence $q \succ p$, which contradicts $p \in \mathcal{P}$. So $\mathcal{P} \subset \widehat{\mathcal{P}}$. Hence $\widehat{\mathcal{P}} = \mathcal{P}$.

A consequence of (9) is that at each step, $\widehat{\mathcal{P}}^n$ is an antichain. Since during the execution of the algorithm all the elements of $S_t$ are compared to all the element of the current $\widehat{\mathcal{P}}$, the algorithm do at most

$$|\mathcal{S}_t| \max_n |\widehat{\mathcal{P}}^n| \leq |\mathcal{S}_t| \text{width}(\mathcal{S}_t)$$

comparisons, and as a consequence

$$\mathbb{P}(\mathcal{E}_1^C) \leq |\mathcal{S}_t| \text{width}(\mathcal{S}_t) \frac{\delta}{|\mathcal{S}_t|^2} \leq \delta.$$

### B.3 Proof of Theorem 4

Now let $\mathcal{R}_1$ be the regret generated by the decoy step. To reach this step, an arm $i$ must be such that $\Delta_i < \varepsilon_{N-1}$. If $i \in \mathcal{P}$, then pulling the arm $i$ produces no regret. Otherwise, it is easy to see that the arm is compared to at most **width**$(S)$ other arms before being eliminated.

$$\mathcal{R}_1 \leq \textbf{width}(S) \log\left(\frac{2NK^2}{\delta}\right) \sum_{i, \Delta_i < \varepsilon_{N-1}, i \notin \mathcal{P}} \frac{\Delta_i}{\Delta^2}$$

$$\leq \left(\frac{\varepsilon_{N-1}}{\Delta}\right)^2 \textbf{width}(S) \log\left(\frac{2NK^2}{\delta}\right) \sum_{i, \Delta_i < \varepsilon_{N-1}, i \notin \mathcal{P}} \frac{1}{\Delta_i}$$

$$\leq K\textbf{width}(S) \log\left(\frac{2NK^2}{\delta}\right) \sum_{i, \Delta_i < \varepsilon_{N-1}, i \notin \mathcal{P}} \frac{1}{\Delta_i}.$$