[Reviews · NeurIPS 2017]

Reviewer 1



This paper deals with dueling bandits on a partially ordered set in the framework of the best arm identification. The goal of the dealing problem is identification of the set of maximal elements by doing as small number of stochastic pair comparisons as possible. They propose a high-probability epsilon-approximate algorithm and a high-probability exact algorithm, and analyze their sample complexities. They also conduct numerical simulations to demonstrate effectiveness of their algorithms. The readability of this paper should be improved. For example, some subtitles such as ContributionsandProblem statement' are not appropriate for the descriptions following the subtitles. Some notion names such as a social poset and a chain in it are inappropriate because a social poset may not be a poset and a chain in it can contain a circle. Algorithm 1,2,3,4 should be pseudo-codes but do not look like pseudo-codes. Considering the relations among Algorithm 1,2 and 3, Algorithm 2 should be first (Algorithm 1) and Algorithm 1 should be the last (Algorithm 3). I cannot understand why what they call peeling is necessary. Why is it a problem to run UBSRoutine only once with epsilon_N? The authors should explain what merits are there using so called peeling. I think UNchainedBandits can fail to output the Pareto front in the case that a social poset contains an isolated cycle. In that case, UBSRoutine always outputs a set that includes at least one of cycle elements. By reading the author’s feedback, now I understood the effect of peeling. As for social posets with cycles, I also understood but I am still feeling something strange. Their problem setting looks nice for regular posets, but not appropriate for social posets. For example, consider a cycle. Then the Pareto front is an empty set, but its epsilon-approximation can be a (2/K)-sized set for any small epsilon. In fact, UnchainedBandits may output (2/K-1)-sized set. In this case, a lot of subsets are epsilon-approximations of the Pareto front, and I don’t feel happy even if one of them can be obtained.

Reviewer 2



The paper addresses a variant of the dueling bandit problem where certain comparisons are not resolvable by the algorithm. In other words, whereas dueling bandit algorithms deal with tournaments (i.e. fully connected directed graphs), the algorithms in this paper deal with more general directed graphs. The authors offer only vague conceptual reasons for the significance of this new problem setting, however from a practical point of view, this is an extremely important problem that the existing literature does not address at all. More specifically, ties between arms are major stumbling blocks for most dueling bandit algorithms; indeed, many papers in the literature explicitly exclude problems with ties from their analyses. On the other hand, in practice one often has to deal with "virtual ties": these are pairs of arms that could not be distinguished from each other given the number of samples available to the algorithm. A good example to keep in mind is a product manager at Bing or Google who has to pick the best ranker from a pool of a hundred proposed rankers using pairwise interleaved comparisons, but only has access to say %1 of the traffic for a week: if it happens to be that there are certain pairs of rankers in the pool that are so similar to each other that they couldn't be distinguished from each other even using the entirety of the available samples, then for all practical purposes the are ties in the resulting dueling bandit problem. So, our product manager would want an algorithm that would not waste the whole traffic on such virtual ties. The authors propose two algorithms that deal with this problem under two different levels of generality: 1- an algorithm to find an \eps-approximation of the pareto front in the more general social poset setting, which is an extension of the Condorcet assumption in the full tournament case; 2- as algorithm to find the exact pareto front in the much more restrictive poset setting, which is an analogue of the total ordering assumption for regular dueling bandits. I personally find the former much more interesting and important because there is plenty of experimental evidence that transitivity is very unlikely to hold in practice, although the second algorithm could lead to very interesting future work. The authors use a peeling method that uses epochs with exponentially increasing lengths to gradually eliminate arms with smaller and smaller gaps. In the analysis, this has the effect of rendering the regret incurred from most edges in the comparison graph negligible, hence the linear dependence on the gaps in regret bounds. The price that one pays for this simpler proof technique is an extra factor of K, which in the worst case makes the bound of the form O(K^2 log T / \Delta). As a general comment, I think it would be a good idea for the authors to devote more space in the paper to situating the results presented here with respect to the existing dueling bandit literature: I think that would be much more helpful to the reader than the lengthy discussion on decoys, for instance. In particular, there is a very precise lower bound proven for Condorcet dueling bandits in (Komiyama et al, COLT 2015), and I think it's important for the authors to specify the discrepancy between the result in Theorem 1 and this lower bound in the full tournament setting when \eps is smaller than the smallest gap in the preference matrix. Furthermore, given the numerous open questions that arise out of this work, I highly recommend having a lengthier discussion of future work. For instance, there is some recent work on using Thompson Sampling for dueling bandits in (Wu and Liu, NIPS 2016) and it is an interesting question whether or not their methods could be extended to this setting. Moreover, it would be interesting to find out if more refined results for the von Neumann problem, such as those presented in (Balsubramani et al, COLT 2016) can be transported here. Another criticism that I have of the write-up is that in the experiments it's not at all clear how IF2 and RUCB were modified, so please elaborate on the specifics of that.

Reviewer 3



The paper studies an interesting variant of dueling bandit problem, on posets, where pairs-of-arms may or may not be comparable to each other. For this setting, a natural problem is to recover the set of optimal arms, ones which are not dominated by any other arm. That is it is needed to recover the best element from all the chains. The authors introduce a very natural algorithm which is reminiscent of successive elimination. Regret guarantees and total number of comparisons are provided. The second part of the paper deals with introduction of decoys to handle the problem of recovery of the entire pareto frontier. Decoys are essentially bad elements of the set that are dominated by every element in the chain. The algorithms introduced are pretty intuitive and the formulations and problem statement novel. I have just one main complaint. It is not clear to me for what algorithm are Theorem 3 and Theorem 4 applicable. Are they applicable for UBS routine instantiated with Decoy comparison? I found it confusing because the paragraph in lines 240-245 seems to hint at a hybrid approach where one runs Unchained bandits with direct comparison upto a certain time before running with decoy comparison thereafter. It would be great if the authors can clarify this.